# Nasal povidone-iodine implementation for preventing surgical site infections: Perspectives of surgical nurses

Eric N. Hammond[1,2], Nicole Brys[3], Ashley Kates[2,4], Jackson S. Musuuza[2,4], Ambar Haleem[2], Michael L. Bentz[5], Nasia Safdar[2,4]*

1 Institute for Clinical and Translational Research, University of Wisconsin-Madison, Madison, WI, United States of America, 2 Division of Infectious Diseases, Department of Medicine, University of Wisconsin School of Medicine and Public Health, Madison, WI, United States of America, 3 Waisman Center, University of Wisconsin-Madison, Madison, WI, United States of America, 4 William S. Middleton Memorial Veterans Hospital, Madison, WI, United States of America, 5 Departments of Surgery (Division of Plastic and Reconstructive Surgery) and Urology, University of Wisconsin School of Medicine and Public Health, Madison, WI, United States of America

* ns2@medicine.wisc.edu

**Data Availability Statement:** All relevant data are within the paper and its Supporting Information files.

## Abstract

### Introduction

Preoperative nasal decolonization of surgical patients with nasal povidone-iodine (PI) has potential to eliminate pathogenic organisms responsible for surgical site infections. However, data on implementation of PI for quality improvement in clinical practice is limited. The purpose of this study was to evaluate the implementation feasibility, fidelity and acceptability of intranasal PI solution application by surgical nurses using the Integrated Promoting Action on Research Implementation in Health Services (i-PARIHS) conceptual framework.

### Materials and methods

Using the i-PARIHS framework to frame questions and guide interview content areas, we conducted 15 semi-structured interviews of pre- and post-operative care nurses in two facilities. We analyzed the data using deductive content analysis to evaluate nurses' experience and perceptions on preoperative intranasal PI solution decolonization implementation. Open coding was used to analyze the data to ensure all relevant information was captured.

### Results

Each facility adopted a different quality improvement implementation strategy. The mode of facilitation, training, and educational materials provided to the nurses varied by facility. Barriers identified included lack of effective communication, insufficient information and lack of systematic implementation protocol. Action taken to mitigate some of the barriers included a collaboration between the study team and nurses to develop a systematic written protocol. The training assisted nurses to systematically follow the implementation protocol smoothly to ensure PI administration compliance, and to meet the goal of the facilities. Nurses' observations and feedback showed that PI did not cause any adverse effects on patients.

**Funding:** The author(s) received no specific funding for this work.

**Competing interests:** The authors have declared that no competing interests exist.

## Conclusions

We found that PI implementation was feasible and acceptable by nurses and could be extended to other facilities. However further studies are required to ensure standardization of PI application.

## Introduction

Despite tremendous advancements in surgical practice, surgical site infection (SSI) remains a major challenge in healthcare, affecting approximately 500,000 patients annually in the United States [1]. Surgical site infections (SSIs) are associated with prolonged hospital stays and the need for antibiotic therapies that are frequently complex, lengthy and potentially, toxic [2, 3]. Nasal carriage of *Staphylococcus aureus* is found in approximately 30% of patients [4, 5] and greatly increases the risk of SSI [2].

To mitigate the risk of *S. aureus* SSIs, current evidence-based guidelines recommend decolonization of patients through application of nasal mupirocin twice daily in the 5 days preceding surgery. Unfortunately, significant drawbacks are associated with this preventive measure, including potential for an emergence of staphylococcal resistance to mupirocin [6, 7] and mupirocin related adverse reactions [8]. In addition, an out-of-pocket cost of the drug and the inconvenience of having to apply mupirocin ointment multiple days prior to surgery, makes patient compliance difficult [8]. Therefore, there is an urgent need to identify and implement an alternative nasal antimicrobial decolonizing agent that is effective, safe, cost-effective, and not reliant on the patient for optimal implementation. In this regard, intranasal povidone-iodine solution (PI) appears to be a promising alternative to mupirocin as an immediate preoperative option the day of surgery.

Intranasal PI has broad-spectrum antibacterial activity, including against methicillin-susceptible and resistant *S. aureus* and *Pseudomonas* [9]. PI is rapidly bactericidal *in vitro* (within 15 to 20 seconds) with effects on skin lasting for 12 to 14 hours after application, a feature that provides microbial suppression in the immediate postoperative period [10, 11]. Unlike mupirocin that needs to be applied by the patient 5 days prior to surgery, intranasal PI solution can be applied by a nurse 1 hour before the procedure. To date, several studies evaluating intranasal PI solution have demonstrated its effectiveness in reducing SSI [4, 12, 13]. Investigators have also shown that intranasal PI solution is more cost-effective than mupirocin and is acceptable to patients [8, 12, 14]. Given this evidence, hospitals are increasingly interested in implementing PI to reduce surgical site infections among surgical patients. However, little is known about the PI implementation process in healthcare settings. The aim of this study was to evaluate the implementation feasibility, fidelity and acceptability of intranasal PI solution application using the i-PARIHS framework.

## Materials and methods

In this study, we evaluated a quality improvement (QI) implementation of PI in the surgical units of two surgical facilities to identify and eliminate barriers that may impede successful implementation.

### Study design

This was a qualitative study based on deductive content analysis with the aim of evaluating implementation feasibility, fidelity and acceptability of intranasal PI solution application by

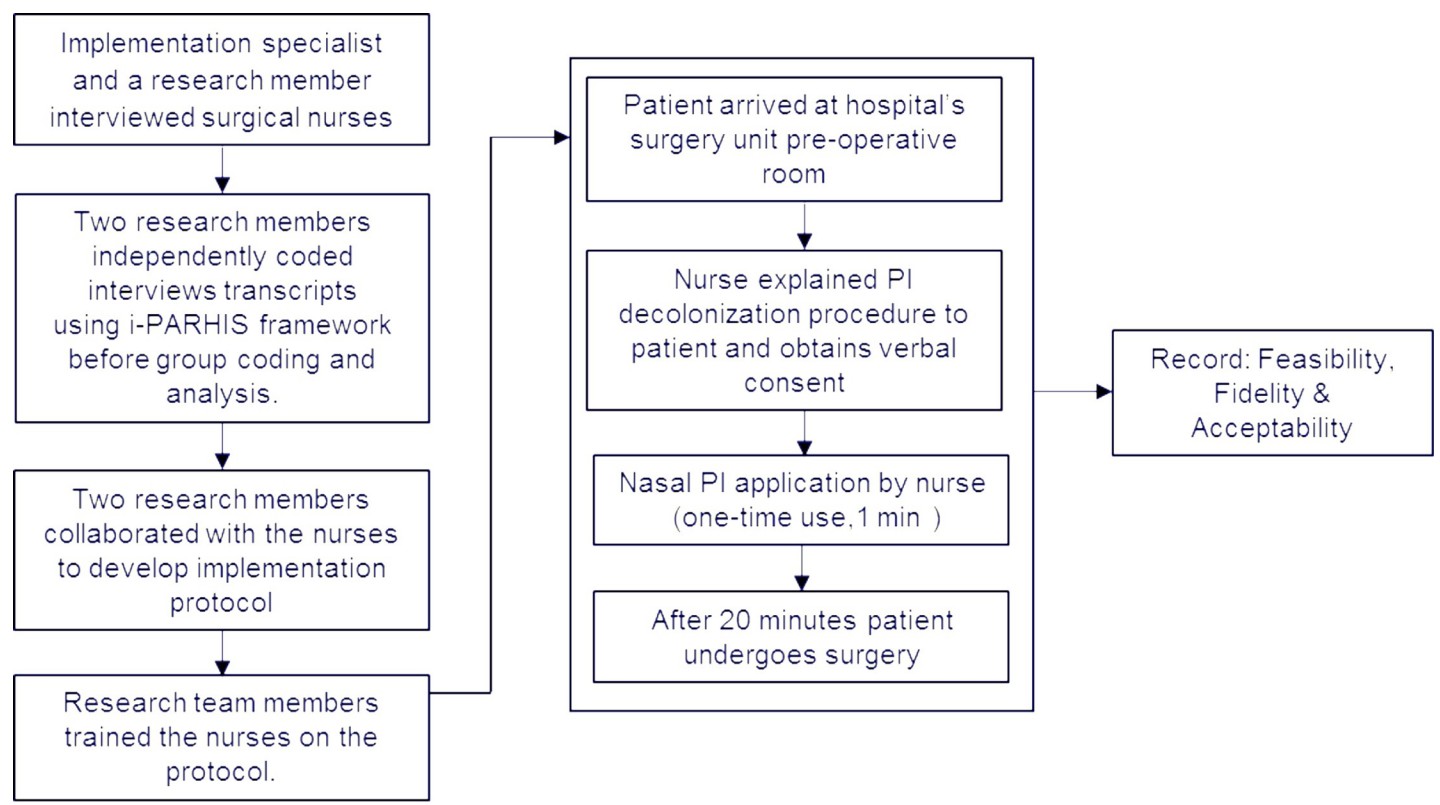

**Fig 1. The schematic diagram of QI study representing pre-implementation process, implementation process and outcome measures.**

nurses in surgery units of two facilities (A and B). A flow diagram representing the QI study is shown in Fig 1.

We used the i-PARIHS framework to evaluate the implementation process [15]. The i-PAR-IHS model is a well-accepted implementation framework outlining constructs relevant to the process of introducing new practices into a healthcare context (such as facilities A and B). This framework focuses on the facilitation activities necessary to support effective implementation. Using the i-PARIHS framework in this QI study, helped document relevant PI implementation processes necessary to determine the most suitable facilitation method to change practice and improve the quality of life of patients.

This framework posits that *facilitation* activates implementation by assessing and responding to the framework's core constructs: the *innovation*, the *recipients*, and the *contextual setting* [15]. For this study, the *innovation* is using nasal PI solution as a pre-surgical decolonization agent. The *context is* facility A and facility B. *Recipients* are the people who are affected by and exert influence on the implementation, this includes patients receiving the nasal PI and the surgical nurses who applied it. Study team facilitated the implementation process by using several commonly accepted implementation strategies [16]. These included leadership mandate: a change in protocol, developing and distributing educational materials, providing training to providers, and providing ongoing consultation to support implementation. The implementation of the pre-surgical nasal PI application protocol was considered successful if providers feasibly implemented the protocol with fidelity in a way that was acceptable to both providers and patients.

## Setting

We implemented this study in an ambulatory surgery center (facility A) and a surgical unit of a hospital (facility B) located in the Midwestern United States. We chose facilities A and B because they were highly engaged sites interested in QI and committed to implementing PI as a novel approach to reducing surgical site infections. Moreover, these facilities have relevant volume of patients presenting with varied surgical cases. These included septoplasty-rhinoplasty–open (plastic surgery involving the nose), blepharoplasty (eyelid surgery), breast reduction, breast augmentation (breast implant) and abdominoplasty (tummy tuck) in facility A. In facility B, these included coronary bypass surgery, heart valve replacement, thoracic surgery, gender affirming procedures, breast cancer and other cancer reconstructive procedures, and extremity salvage procedures including revascularization and skeletal and soft tissue trauma reconstruction.

## Preparation and logistics

The study team met with the quality improvement committee members for facilities A and B. This committee comprised two clinical infection control practitioners, an administrative director, two nurse managers and a quality officer. The study team also comprised an implementation specialist, a surgeon, two infectious diseases specialists and two microbiologists. The study team obtained approval from the committee during the meeting to interview the pre- and post-surgical nurses directly involved in the implementation. Two members of the study team later had a site visit tour at the two facilities. The nurse manager from each facility introduced the study team to the pre- and post-surgical nurses on duty to initiate and establish rapport prior to interviews.

## Existing standard decolonization procedures before surgery at the study facilities

On the night before and morning of surgery, each patient received chlorhexidine (CHG) wipes or CHG (Hibiclens® 4%) bath at home. Patients who were taken urgently to surgery were given either a one–time CHG bath or no-rinse wipes.

Each participant subsequently received intravenous antimicrobial prophylaxis: cefazolin or clindamycin if there is allergy to the former. Patient colonized with Methicillin-resistant *Staphylococcus aureus* received vancomycin 15mg/kg in addition to cefazolin and a 5-day nasal decolonization with mupirocin nasal.

## Pre-operative PI intervention at the study facilities

Facilities A and B used nasal povidone-iodine solution 10%w/w (Professional Disposables International, Inc, Orangeburg, NY) as the nasal decolonizing agent to prevent potential SSI. The kit comprised of 4 pre-saturated swab sticks with 10% w/w povidone-iodine antiseptic solution.

## Nurses' communication with patients about pre-operative PI intervention

Prior to surgery, nurses explained to patients about efforts to reduce the risk of infections after surgery and hence the need for nasal application an antiseptic (povidone-iodine) before surgery. Patients were informed that the antiseptic would be directed against nasal germs that could cause a surgical site infection. Patients were also told about the possibly of adverse events of this antiseptic application such as leaving a temporary brown stain inside the nose, potential allergic reactions and what to do if they noticed any of these. For patients with a known PI allergy or those not consenting to the nasal application, they were offered an alternative existing decolonization protocol instead.

**Patients inclusion and exclusion criteria.** The nurses administered PI to the nares of all patients who underwent surgical procedures in facilities A and B. Patients with known allergies to iodine or those scheduled for surgical procedures which prohibited them from receiving PI were excluded.

## PI application

For all surgical patients, a trained surgical nurse carefully inserted 1–2 cm of a PI-saturated swab into their anterior naris on the first side and gently moved it in a circular motion along the nasal wall (covering all surfaces) for at least 15 seconds. This process was repeated on the same anterior naris with a second unused swab and similarly on the second naris with 2 new swabs. In total PI application time was 60 seconds for each patient (30 seconds per naris). For facility A, patients waited for at least 20 minutes before surgery. There was no specific waiting time for facility B. The PI was applied according to the manufacturer's instructions.

## Participant recruitment

We used several recruitments strategies, which included unit emails, announcing the QI study during unit meetings and e-mail invitations. Leaders from the surgery units informed the surgical staff via email letting them know that an implementation specialist would be inviting them to participate in interviews about PI implementation and that participation was voluntary.

The implementation specialist obtained a list of nurses' names from nurse managers and followed up individually with the surgical staff via email. The nurses were informed that interviews would be in-person face-to face, audio recorded, and transcribed with all identifiers removed.

## Participants

The sample population consisted of 15 pre- and post-surgical nurses employed in facilities A and B. This comprised of 5 surgical nurses working at facility A and 10 surgical nurses working at facility B. We chose them because they had experience in PI application and could provide in-depth knowledge of current PI procedures in place. Other surgical staff who were involved in the PI implementation process, but not directly involved in PI application were excluded. We used convenient sampling to sample the nurses.

## Data collection

Interviews were conducted with nurses at facilities A and B in May and November 2019, respectively. We used a semi-structured interview guide (S1 File) to conduct in-person (face-to-face) interviews with the surgical nurses (n = 15). Each interview lasted approximately 20–30 minutes in a private office in facilities A and B. We used open-ended questions to assess the surgical nurses' experiences of nasal implementation of PI in their facilities, such as, having them outline the actual process of preoperative PI application. We also inquired about challenges that the nurses encountered with the PI application process, including any factors that may have prevented them from completing the process, such as patient discomfort or development of adverse effects.

The interviews were audio-recorded and transcribed verbatim. We frequently reviewed and updated the interview questions based on previous responses. This assisted us to shape our questions to meet the objective of the study. We also took field notes to ensure all necessary information that could not be captured in the audio-recording device was documented. This

included observations, facial expression, gestures and other relevant information. We used Dedoose v 8.0.35 [17] a qualitative data analysis software to manage our data.

In order to minimize breach of confidentiality, we only obtained personal information relevant to the study. Furthermore, only the investigators and authorized study staff had access to the data. All interviewees' data were handled confidentially and kept securely.

## Outcome measures

We analyzed the implementation using the 3 outcome measures to determine if the implementation was successful. (1) Feasibility of implementing the pre-surgical nasal PI application protocol in the surgical setting. The purpose of evaluating the PI implementation feasibility was to determine if the implementation can work in these facilities. To assess this, we examined the barriers encountered by surgical nurses while implementing the protocol and whether it was possible to adequately deal with these barriers. (2) Fidelity or whether the pre-surgical nasal PI intervention was delivered as intended. This was assessed by asking providers how frequently they were unable to deliver the intervention exactly as described in the protocol and by looking at whether there were patients who were eligible to receive the intervention during the implementation period that did not receive it. (3) Acceptability of the intervention to providers (nurses) and patients. We assessed this by asking nurses about any burden caused by the intervention and the reactions of patients to the application of the nasal PI solution.

## Data analysis

Data were analyzed by study team members who were skillful and knowledgeable in deductive content analysis. Two team members were involved in coding and analyzing the interview transcripts using deductive content analysis [18]. Data were reviewed for content corresponding to the i-PARHIS framework component as described above [15].

Each team member reviewed the interview transcripts and coded independently. The two team members later reconvened to discuss and compare the coding process. The group reviewed each transcript line-by-line (open coding) together. This was to ensure all relevant information was captured.

Discrepancies in the coding and analysis were discussed between the two reviewers until a consensus was reached. This also helped to reduce individual bias. The transcripts from participants' interviews were coded in Dedoose [17]. Supportive interview quotes from participants were added as evidence to emphasize a specific point.

## Ethical approval

The University of Wisconsin Health Sciences Institutional Review Board exempted this study as a quality improvement project, and we obtained informed verbal consent from each participant.

## Results

All 5 pre- and post-nurses responsible for decolonizing surgery patients with PI in facility A participated in the interview while 10 out 11 pre- and post-nurses in facility B also participated. Nurses at facility A had worked there between 5 months and 7 years, whereas nurses at facility B had worked at that site between 6 months and 18 years. Participating nurses explained their responsibilities as preparing, admitting, and caring for pre- and post-surgery patients. Each nurse confirmed their roles and experience in the PI implementation. They also explained how the QI implementation was handled, described PI decolonization process and rated the general implementation process. At facility A, PI was always available with the exception of one

occasion where there was shortage of it. PI was available consistently at facility B. Implementation strategies and nurse training at facilities A and B varied. Whereas, in-person training of PI application was conducted at facility A, nurses at facility B were provided online training.

## Context

In this study, the context focused on the role of management in PI implementation. In both facilities interviewees indicated that management of the facilities were fully invested making the PI implementation process a success. Nurses explained that they were informed about the PI implementation during weekly unit meetings and via email. The facility A nurses did point out that there was a short interval between when they were informed about the PI implementation and the date that the initiative was launched.

> ". . .. it was just pretty quick. . . . we were told we were going to start doing it, and then, or we were told that we were supposed to start doing it, and then I think we got the training like a day later. . ."(Interviewee 3).

A nurse from facility A stated that there was a prepared protocol but that was not clear enough and lacked important information. Other nurses reported that they only received brief verbal instructions on PI implementation.

> ". . .. I had looked up online the studies and why we would be doing it. We were just basically told, this is what we're going to do. We weren't really told exactly why or what the studies were or anything. We were just told this is what we're going to do. So, I went online and looked it up myself. . ." (Interviewee 4).

> ". . .. We had somebody come and talk to us a little about, like a rep come and talk. And we got, I think we got an email or something saying that we were going to be doing this, and the rep was coming. . . ., she showed us to do stuff, but there wasn't really a whole lot of information, . . . we had a lot of questions from our patients, like how long does this last, or what does, what are the studies showing, and all that . . ., I don't think we had any of that information. . .[*sic*]." (Interviewee 1).

According to nurses at facility B, they were provided with adequate sensitization to prepare them for implementation of preoperative PI decolonization. In addition, PI nasal application training was incorporated into their unit's computer-based training (CBT). A nurse from facility B expounded:

> ". . . First, we heard about it in one of our huddles, and then I'm pretty sure, yeah, there was a CBT, I think, that we had to go through, and there's some videos. And then like a month later, then it finally came out, so we had to go back and look at the videos again and make sure. But, yeah, it was pretty good. . ." (Interviewee 10).

When each nurse was asked to rate the PI implementation process on a scale of 1 (poor) to 10 (excellent), nurses from facility A gave an average of 6.8 (range between 6 and 8). Facility B nurses rated the whole process at an average of 8.4 (range between 7 between 10).

## Recipients

From each unit, the nurses obtained the PI and administered it to the patients prior to surgery. The nurses reported that the PI implementation process was easy and quick once they had

received appropriate information and training and not as complicated or time consuming as they may have initially expected. Generally, all the nurse supported the idea of incorporating into the pre-surgical workflow PI implementation to reduce SSI.

A few nurses from both facilities explained:

". . . I think it's a good treatment, and it's a good idea to do it. . . . if we keep people from getting any sort of infections. I know infections are expensive. So, I think it's important, and they could expand it to other surgeries too, because it's not like it takes you a long time to do it or anything like that. It goes fast . . ." (Interviewee 7).

". . .if we're preventing the spread of MRSA into incisions . . . that's a good thing, because nobody wants to get sicker . . ." (Interviewee 11).

The nurses from facility A shared their frustration about the lack of detailed information. They explained that having adequate information would have increased their knowledge and ensured consistency.

". . .That everybody is doing the same thing, . . .. Because if you just hear it once verbally, it's different than you see it written down. . ." (Interviewee 3).

At both facilities, patients accepted idea of preoperative intranasal PI application to reduce SSI once nurses had clearly explained the relevance of the implementation. All patients underwent PI treatment with no refusals.

## Facilitator

In i-PARIHS framework, a facilitator is responsible for planning and guiding implementation process. The facilitator could either be from the facility setting (inner) or outside (outer). Both inner and external facilitators can also coordinate together to ensure implementation. At facility A, an internal facilitator gave a verbal instruction to the nurse to carry out PI application. Barriers encountered by the nurses included limited information and training and consequently, inconsistency in practice. An external facilitator, PI representative demonstrated PI application to the nurses. To some extent, this provided the nurses with additional information on PI application, however it was not adequate. Some nurses were unsure if their PI implementation technique was consistent with the intended purpose.

". . . She (Povidone-iodine representative) came and showed us the product, how to open everything, how long to use it, where, you know, the right way to use, I guess, how to put it in the nose and how far to go in the nose and how long we had to do it . . .... ." (Interviewee 1).

The study team successfully developed the PI application protocol in consultation with the nurses. The iterative feedback, suggestions and comments by the nurses was critical in developing the protocol. The final protocol was reviewed, corrected, and accepted by the nurses before it was adopted by the facility. The nurses were able to follow the protocol systematically in order to effectively perform their task. The protocol was clear and easy to follow. Corrective action implemented during the implementation was effective to eliminate gaps and barriers. One of the nurses stated that she was excited that we were seeking their input in the implementation of PI initiative.

". . .I'm surprised, honestly, you guys are even asking us our opinion. . ." (Interviewee 3).

At facility B, facilitation was via online resources. Nurses watched a demonstration of PI application on You-tube. Although the training link was always available to them, there was no in-person facilitator. According to the nurses, the training mode was successful.

Written information provided to nurses allowed them to explain the significances of PI application more clearly to their patients. Based on the information received during training they were able to administer PI to patients. One nurse did indicate that she forgot some of the steps of the implementation process at times due to a lack of a written protocol.

### Innovation

Both facility A and B nurses implemented a novel pre-operative nasal decolonization intervention using PI. Since PI was new to these facilities, nurses underwent training to enable them to administer the treatment to their patients exactly as described in the protocol. Nurses and patients acknowledged that PI application was a novel strategy to that could improve the long-term health of surgical patients. All the nurses indicated that PI application was easy to use, and that it took them less than 5 minutes to explain and carry out the procedure.

In general, no major adverse effects were observed of the PI treatment. Of the 15 nurses interviewed, 10 reported that patients had mentioned that PI trickled down patients' throat and caused a bad taste in their mouth.

". . .. Lots of people said it (PI) burned, made them feel like their nose was running, smells. . .. Sometimes it ran in the back of their throat, . . .. they were always like, oh, it goes away, and it's fine . . ." (Interviewee 1).

### Discussion

In this study, we explored implementation feasibility, fidelity and acceptability of preoperative intranasal PI solution to reduce surgical site infections using the i-PARIHS conceptual framework. We found that the role of an implementation facilitator is critical to the success of a QI initiative. This is consistent with findings from a study conducted on the impact of facilitator in QI implementation in healthcare setting [19]. In the i-PARIHS framework, the facilitator develops activities that effectively and directly guide the implementers (nurses) to achieve the aim(s) of the project. The facilitator also provides information on the implementation process, via various trainings modalities, workshops, and a systematic written protocol. Our study confirmed that the more information the nurses had, the easier it was for them to educate their patients about the implementation process.

Interviews with nurses from facility A revealed there was insufficient information at the pre-implementation stage which resulted in the nurses not being able to fully understand the process. Therefore, it was difficult for the facility A nurses to respond to patient's questions related to PI which partly accounted for nurses' frustration with the process.

In contrast, nurses at facility B reported that they received adequate information during the pre-implementation training and that additional information was provided to enhance their understanding and equip them with knowledge to answer patients' questions after the online training. This accounts for the high implementation process rating (between 7–10) among nurses from facility B.

Another barrier that impacted QI implementation was short notification period. We noted that QI implementation information to nurses at facility A was rushed which resulted in inadequate preparation. At facility B, nurses were continually informed for at least 4 weeks prior to the implementation. As part of the preparation strategy, nurses were also enrolled in an online

PI implementation training during their regular Computer-based training (CBT). This was to ensure that the nurses were fully equipped to implement the implementation. Additionally, the training link was available online for nurses to watch repeatedly at their convenience. Repetition plays an essential role in implementation stages because it provides the nurses with extra support to learn, rehearse and improve upon their skills to execute the protocol as intended to be [20]. This also explained why the nurses at facility B appeared more prepared than those at facility A for the implementation. This approach highlights the importance of detailed planning and preparation at the pre-implementation stage of any new and novel initiative.

An additional barrier observed was lack of effective communication. Effective communication enhances quality and promotes safety [21]. Data obtained showed that whereas at facility A nurses were given verbal instructions on how to administer PI, at facility B, information was provided via an online video. Neither of these facilities had a written protocol for implementation. This increased the potential for noncompliance among nurses. Having access to a step-by-step written protocol is crucial since it affords the nurses a reference in case of uncertainties or questions about the implementation process.

In instances where poor communication and instruction exist, there is a high probability of the occurrence of serious errors [22, 23]. This can jeopardize a QI initiative and could affect the effectiveness of the intervention on patients. Moreover, poor communication can mislead nurses to administer intervention inaccurately and harm patients. Procedures involved in implementation must be clearly written down and communicated efficiently with nurses.

## Corrective actions to improve the implementation process

To effectively address the barriers encountered in PI implementation, the team collaborated with the pre- and post-surgical nurses at facility A to develop a PI application protocol specifically for that facility. The team drafted the PI protocol based on the baseline data collected and shared it with the nurses to review. The nurses reviewed and made valuable suggestions which were incorporated into the updated version. The protocol was developed until the nurses and study team deemed it fit to be implemented. Once the protocol was approved, the study team facilitated the protocol training with all five nurses in attendance. We continued to modify the protocol even after the initiation of the implementation process which was necessary to allow adaptability of the protocol at facility A.

The ability of the nurses to adhere to the protocol and administer the intervention is critical [24] to the success of a QI process. Therefore, the study team trained all nurses to follow the protocol step-by-step in order to administer the intervention (Fig 1) to ensure high fidelity is achieved. During the training, frequent questions asked by patients were incorporated and addressed in the protocol. This increased the knowledge and confidence of the nurses and eliminated the frustrations that they had reported described in their interviews. We also made the finalized protocol available to nurses for reference and as a future training tool to ensure standardization.

The QI implementation protocol training highlights included: (1) the purpose of preoperative nasal decolonization with PI; (2) patient preparation and education; (3) inclusion and exclusion of patients; (4) use of personal protective equipment; (5) opening of PI tube; (6) application of PI to patient's nares and (7) PI allergy management.

## Implementation outcomes

We further evaluated the success of the PI implementation process by using three implementation indicators—fidelity, feasibility and acceptability [25, 26]. Despite the multiple barriers

encountered during the implementation process, the nurses reported that PI application was easy and straightforward, that patient education and decolonization process itself did not pose a constraint on their time. Additionally, PI implementation did not negatively affect them or their work.

Nurses' observations and general feedback showed that the PI solution did not cause any major adverse effects on patients, except for a residual brown discoloration in their nose which could be washed off immediately with water or by patient blowing their nose after surgery. In contrast, Philips et al. [4] recorded that 16 out of 842 patients who received PI had experienced adverse effects and were excluded from their randomized controlled trial. Among the group that experienced adverse effects, one patient experienced vasovagal reaction during the PI application. In summary, information collated from the i-PARIHS consult illustrated that PI implementation was feasible and had a high acceptability and fidelity.

## Lessons learned

For this study we learned that:

1. The active involvement of facility management (leadership or champion) in implementation is critical. Therefore, there is a need for the study team to obtain leadership buy-in prior to the initiation of the study.

2. Adequate time must be devoted by management to systematically sensitize and train all nurses particularly those involved in the QI implementation.

3. The nurses or direct users of the protocol must always be involved in its development. By doing so, they would be able to better understand the process, be more involved and contribute positively to ensure consistency and accuracy of implementation.

4. Once the QI implementation protocol is finalized, it should be made accessible to all users during the implementation process.

This paper has several limitations: First, the QI implementation strategy adopted was unique to each facility in this study, therefore, results from this study may not be generalizable. Secondly, nurses' feedback on the implementation process from the two facilities may differ from other surgical units. Thirdly, at the pre-implementation stage, the nurses seemed unfamiliar with the differences between QI and human research studies. Through discussion, the study team clarified the distinction between the concepts prior to the initiation of the study. Finally, during the implementation, we were unable to observe if nurses were fully able to translate the online training into practice as stated in the protocol.

## Conclusions

Data obtained from this study suggested that the PI quality improvement intervention implemented in facilities A and B was feasible and acceptable by nurses and patients. Furthermore, incorporating i-PARIHS in this QI study allowed us to document all relevant PI implementation processes that enabled management of the facilities to determine the most suitable facilitation method to change practice and improve quality of life of patients. To achieve a successful QI implementation, active management involvement, facilitator, suitable participants, adequate time for preparation, effective planning and availability of educational materials are paramount. Further studies are required to ensure standardization of the PI implementation and to explore the perspectives of all stakeholders, including the patient.

## Supporting information

**S1 File. Interview guide for nurses to evaluate pre-surgical nasal povidone-iodine implementation.**
(PDF)

## Acknowledgments

The authors thank all participants and unit staff for sharing their experience to strengthen this study. We would also like to thank Ms. Michele Zimbric, Division of Infectious Diseases, Department of Medicine, University of Wisconsin School of Medicine and Public Health, Madison for her assistance during the study.

## Author Contributions

**Conceptualization:** Eric N. Hammond, Nicole Brys, Nasia Safdar.

**Formal analysis:** Eric N. Hammond, Nicole Brys, Ashley Kates, Jackson S. Musuuza, Ambar Haleem, Michael L. Bentz, Nasia Safdar.

**Methodology:** Eric N. Hammond, Nicole Brys, Ashley Kates, Jackson S. Musuuza.

**Project administration:** Eric N. Hammond, Nicole Brys, Ashley Kates.

**Resources:** Eric N. Hammond, Nicole Brys, Ashley Kates, Ambar Haleem.

**Software:** Eric N. Hammond, Nicole Brys, Jackson S. Musuuza.

**Supervision:** Ambar Haleem, Michael L. Bentz, Nasia Safdar.

**Writing – original draft:** Eric N. Hammond, Nicole Brys, Ashley Kates, Jackson S. Musuuza, Ambar Haleem, Michael L. Bentz, Nasia Safdar.

**Writing – review & editing:** Eric N. Hammond, Nicole Brys, Ashley Kates, Jackson S. Musuuza, Ambar Haleem, Michael L. Bentz, Nasia Safdar.

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
