## [Decision Letter · Decision Letter 0]

17 Sep 2020

PONE-D-20-21321

Nasal povidone-iodine implementation for preventing surgical site infections: Perspective s of surgical nurses

PLOS ONE

Dear Dr. Hammond,

Thank you for submitting your manuscript to PLOS ONE. After careful consideration, we feel that it has merit but does not fully meet PLOS ONE’s publication criteria as it currently stands. Therefore, we invite you to submit a revised version of the manuscript that addresses the points raised during the review process.

Comments from two anonymous reviewers have been included below for your consideration in revision of your submission.

We look forward to receiving your revised manuscript.

Kind regards,

Kourosh Parham, Ph.D., M.D.

Academic Editor

PLOS ONE

Reviewers' comments:

Reviewer's Responses to Questions

**Comments to the Author**

1. Is the manuscript technically sound, and do the data support the conclusions?

Reviewer #1: Yes

2. Has the statistical analysis been performed appropriately and rigorously? 

Reviewer #1: Yes

3. Have the authors made all data underlying the findings in their manuscript fully available?

Reviewer #1: Yes

4. Is the manuscript presented in an intelligible fashion and written in standard English?

Reviewer #1: Yes

5. Review Comments to the Author

Reviewer #1: This is an interesting study examining the actual implementation of povidone-iodine as pre-operative nasal decontamination.

I appreciated the settings of the study in both a hospital and outpatient surgery center. I recommend including information as to the types of procedures before which povidone iodine was implemented. This may be variable, but if it was mostly orthopedic cases, for example, this should be included.

Please provide more information as to if every patient at this facility receives a MRSA swab.

Please include a brief summary of risk discussion held with patients. Within this, a brief comment should be made regarding ciliotoxicity of 10% PVP-I used intranasally. The article listed below demonstrates ciliotoxicity at 2.5% and above. I would recommend referencing exploration of lower dose concentrations, however not vital to the study at hand.

Reimer K, Wichelhaus TA, Schafer V. Antimicrobial effectiveness of povidone-iodine and consequences for new application areas. Dermatology 2002;204 Suppl 1:114-20.

I recommend making a mention for further studies to be conducted exploring the patient’s perspective as well.

In the discussion where you discuss that nurses could watch videos at convenience, I would recommend touching on the fact that repetition was important during the implementation stage as well rather than just pointing out the availability of online material during the pre-implementation stage.

In the implementation outcomes section, I don’t think that it’s accurate to assume that increased adverse effects in the cited study were based on mode of application or manufacturer. This study most likely had many more patients included and therefore noted these side effects.

Lastly, the lessons learned section was very well done.

6. PLOS authors have the option to publish the peer review history of their article (what does this mean?). If published, this will include your full peer review and any attached files.

Reviewer #1: No

Reviewer 2 Comments:

This study examined the preoperative nasal decontamination using povidone-iodine used by the nursing staff. The study setting for the use of nasal decontamination by PVPI has been greatly demonstrated by the authors for both inpatient and outpatient setting. I would have liked to see the type of preoperative discussion that was performed with patients about the risk of using intra nasal decontamination swabs including the ciliotoxic effects of 10% PVPI and the type of cases that these preoperative procedures were performed.

The authors discussed their implementation with nurses watching videos which makes the process easy. It would be important to include what limitations were in implementation stage as well as pre-implementation stage.

Overall this was a well thought out and implemented study.

---

## [Author Response · Author response to Decision Letter 0]

25 Oct 2020

RE: PONE-D-20-21321:

Dear Dr. Parham,

We thank you and the reviewers for the careful review and thoughtful feedback on our manuscript, “PONE-D-20-21321: Nasal povidone-iodine implementation for preventing surgical site infections: Perspectives of surgical nurses.” 

We have revised the manuscript according to the comments and believe that it is substantially improved with the incorporation of these edits. Below, we provide a point-by-point reply to reviewers’ comments. We have included a marked copy of the revised manuscript that highlights changes, as well as a version without tracked changes. We have also ensured that our manuscript meets PLOS ONE's style requirements.

Thank you for your consideration of our revised manuscript. 

Reviewer #1 comments

Reviewer comment: This is an interesting study examining the actual implementation of povidone-iodine as pre-operative nasal decontamination.

Authors’ reply: We thank the reviewer for this comment recognizing the importance of our study.

In the introduction section, we have inserted “and is acceptable to patients.” (Page 4, Line 74).

Reviewer comment: I appreciated the settings of the study in both a hospital and outpatient surgery center. I recommend including information as to the types of procedures before which povidone iodine was implemented. This may be variable, but if it was mostly orthopedic cases, for example, this should be included.

Authors’ reply: We agree with the reviewer’s comment. We have revised the setting section and added the types of surgical procedures performed on patients who received povidone-iodine application before surgery (Page 5, Lines 117 - 120).

 “These included septoplasty-rhinoplasty – open (plastic surgery involving the nose), blepharoplasty (eyelid surgery), breast reduction, breast augmentation (breast implant) and abdominoplasty (tummy tuck) in facility A. In facility B, these include coronary bypass surgery, heart valve replacement, thoracic surgery, gender affirming procedures, breast cancer and other cancer reconstructive procedures, and extremity salvage procedures including revascularization and skeletal and soft tissue trauma reconstruction.” (Page 5, Lines 117 -120).

Reviewer comment: Please provide more information as to if every patient at this facility receives a MRSA swab.

Authors’ reply: We have revised the” Pre-operative PI intervention at the study facilities” section. We have inserted patient’s inclusion and exclusion criteria which the nurses used to access patient’s eligibility for povidone-iodine administration (Page 7, Lines 154 -157).

 “Patients inclusion and exclusion criteria: The nurses administered PI to the nares of all patients who underwent surgical procedures in facilities A and B. Patients with known allergies to iodine or those scheduled for surgical procedures which prohibited them from receiving PI were excluded.” (Page 7, Lines 154 -157).

Reviewer comment: Please include a brief summary of risk discussion held with patients. Within this, a brief comment should be made regarding ciliotoxicity of 10% PVP-I used intranasally. The article listed below demonstrates ciliotoxicity at 2.5% and above. I would recommend referencing exploration of lower dose concentrations, however not vital to the study at hand.

Reimer K, Wichelhaus TA, Schafer V. Antimicrobial effectiveness of povidone-iodine and consequences for new application areas. Dermatology 2002;204 Suppl 1:114-20

Authors’ reply: We have revised the “Pre-operative PI intervention at the study facilities” section and inserted a brief summary of discussion that was held between nurses and patients under subtitle “Nurses’ communication with patients about pre-operative PI intervention”. We also included the potential risk of povidone-iodine that was discussed with patients (Page 7 lines 144 - 153). 

Prior to surgery, nurses explained to patients about efforts to reduce the risk of infections after surgery and hence the need for nasal application an antiseptic (povidone-iodine) before surgery. 

Patients were informed that the antiseptic would be directed against nasal germs that could cause a surgical site infection. Patients were also told about the possibly of adverse events of this antiseptic application such as leaving a temporary brown stain inside the nose, potential allergic reactions and what to do if they noticed any of these. For patients with a known PI allergy or those not consenting to the nasal application, they were offered an alternative existing decolonization protocol instead. (Page 7, Lines 144 -153).

We have inserted a subtitle “PI application” (Page 7, Line 158).

Reviewer comment: I recommend making a mention for further studies to be conducted exploring the patient’s perspective as well.

Authors’ reply: We have revised our conclusion section and deleted, “Since only two surgical units were involved in this study, further studies are required to ensure standardization of the PI implementation.”.

We have inserted “Further studies are required to ensure standardization of the PI implementation and to explore the perspectives of all stakeholders, including the patient.” (Page 20, Line 443 - 444). 

Reviewer comment: In the discussion where you discuss that nurses could watch videos at convenience, I would recommend touching on the fact that repetition was important during the implementation stage as well rather than just pointing out the availability of online material during the pre-implementation stage.

Authors’ reply: We have revised our discussion section. We inserted “repeatedly” (Page 16, Line 362).

We have inserted “Repetition plays an essential role in implementation stages because it provides the nurses with extra support to learn, rehearse and improve upon their skills to execute the protocol as intended to be[20].” (Page 16, Lines 362-364).

We have added one more citation “[20].”to support our argument that repetition plays important role in implementation stages. 

We have updated citations of the manuscript. 

Reviewer comment: In the implementation outcomes section, I don’t think that it’s accurate to assume that increased adverse effects in the cited study were based on mode of application or manufacturer. This study most likely had many more patients included and therefore noted these side effects.

Authors’ reply: We deleted “in” and replaced it with “from”. We changed “control” to controlled’. (Page 18, Line 411).

In line 411-413; we inserted “Among the group that experienced adverse effects, one patient experienced vasovagal reaction during the PI application.” (Page 18)

We agree with reviewer’s comment. After we have carefully analyzed the reviewer’s comment, we decided to delete, “The increased number of participants with adverse effects, may be due to the mode of PI application or different PI manufacture.” This is because the participants in the povidone-iodine group who experienced adverse effect and were discontinued from the study may be due to other factors instead of mode of PI application or manufacturer or large sample size. (Page 18, Lines 410 -413).

Reviewer #1 comment: Lastly, the lessons learned section was very well done.

Authors’ reply: We thank the reviewer for this comment.

Reviewer #2 comments

Reviewer comment: This study examined the preoperative nasal decontamination using povidone-iodine used by the nursing staff. The study setting for the use of nasal decontamination by PVPI has been greatly demonstrated by the authors for both inpatient and outpatient setting. I would have liked to see the type of preoperative discussion that was performed with patients about the risk of using intra nasal decontamination swabs including the ciliotoxic effects of 10% PVPI and the type of cases that these preoperative procedures were performed.

Authors’ reply: We have addressed this comment in reviewer #1 comment above. Please see reviewer #1 comment (Page 7, Line 144 -153).

Reviewer comment: The authors discussed their implementation with nurses watching videos which makes the process easy. 

It would be important to include what limitations were in implementation stage as well as pre-implementation stage. 

Authors’ reply: We have revised the limitation section and added the pre-implementation and implementation limitations. (Page 19, Lines 427 - 434).

We deleted “(1)’ and inserted “First” (Page, 19, Line 427)

We deleted “(2)” and inserted “Secondly” (Page 19, Line 428)

We also added “Thirdly, at the pre-implementation stage, the nurses seemed unfamiliar with the differences between QI and human research studies. Through discussion, the study team clarified the distinction between the concepts prior to the initiation of the study. Finally, during the implementation, we were unable to observe if nurses were fully able to translate the online training into practice as stated in the protocol.” (Page 19, Lines 430 – 434)

Reviewer comment: Overall, this was a well thought out and implemented study.

Authors’ reply: We thank the reviewer for this comment.

Sincerely, 

Nasia Safdar, MD, PhD.

Vice-Chair for Research, Department of Medicine

Infectious Diseases, University of Wisconsin-Madison. 

Email: ns2@medicine.wisc.edu.

---

## [Editor Report · Decision Letter 1]

29 Oct 2020

Nasal povidone-iodine implementation for preventing surgical site infections: Perspectives of surgical nurses

PONE-D-20-21321R1

Dear Dr. Hammond,

We’re pleased to inform you that your manuscript has been judged scientifically suitable for publication and will be formally accepted for publication once it meets all outstanding technical requirements.

Kind regards,

Kourosh Parham, Ph.D., M.D.

Academic Editor

PLOS ONE

Additional Editor Comments (optional):

I thank you for your positive responses to the reviewers' comments.

---

## [Editor Report · Acceptance letter]

9 Nov 2020

PONE-D-20-21321R1 

Nasal povidone-iodine implementation for preventing surgical site infections: Perspectives of surgical nurses 

Dear Dr. Hammond:

I'm pleased to inform you that your manuscript has been deemed suitable for publication in PLOS ONE. Congratulations! Your manuscript is now with our production department. 

Kind regards, 

on behalf of

Dr. Kourosh Parham 

Academic Editor

PLOS ONE